# Krill Protein Hydrolysate Provides High Absorption Rate for All Essential Amino Acids—A Randomized Control Cross-Over Trial

**DOI:** 10.3390/nu13093187

**Published:** 2021-09-14

**Authors:** Rebekka Thøgersen, Hanne Christine Bertram, Mathias T. Vangsoe, Mette Hansen

**Affiliations:** 1Department of Food Science, Aarhus University, Agro Food Park 48, 8200 Aarhus N, Denmark; rebekka.thoegersen@food.au.dk (R.T.); hannec.bertram@food.au.dk (H.C.B.); 2Aker Biomarine Antarctic AS, Oksenøyveien 10, NO-1327 Lysaker, Norway; mathiasvangsoe@gmail.com; 3Department for Public Health, Aarhus University, Dalgas Av. 4, 8000 Aarhus C, Denmark

**Keywords:** krill protein, protein hydrolysate, amino acid, NMR spectroscopy, whey, soy, human, marine protein, protein accessibility, postprandial absorption, alternative proteins

## Abstract

Background: adequate protein intake is essential to humans and, since the global demand for protein-containing foods is increasing, identifying new high-quality protein sources is needed. In this study, we investigated the acute postprandial bioavailability of amino acids (AAs) from a krill protein hydrolysate compared to a soy and a whey protein isolate. Methods: the study was a randomized, placebo-controlled crossover trial including ten healthy young males. On four non-consecutive days, volunteers consumed water or one of three protein-matched supplements: whey protein isolate, soy protein isolate or krill protein hydrolysate. Blood samples were collected prior to and until 180 min after consumption. Serum postprandial AA concentrations were determined using ^1^H NMR spectroscopy. Hunger and satiety were assessed using visual analogue scales (VAS). Results: whey and krill resulted in significantly higher AA concentrations compared to soy between 20–60 min and 20–40 min after consumption, respectively. Area under the curve (AUC) analyses revealed that whey resulted in the highest postprandial serum concentrations of essential AAs (EAAs) and branched chain AAs (BCAAs), followed by krill and soy, respectively. Conclusions: krill protein hydrolysate increases postprandial serum EAA and BCAA concentrations in a superior manner to soy protein isolate and thus might represent a promising future protein source in human nutrition.

## 1. Introduction

Adequate protein intake is essential for human health, well-being and performance, since the body needs protein to build and repair tissue, to form muscle and for a vast array of cellular functions [1].

Nowadays, there is a growing demand for dietary protein. The United Nations Food and Agriculture Organization (FAO) has estimated that the world population will reach 9.8 billion people by 2050, which means that global food production must increase more than 70% to feed everyone [2]. Furthermore, within the sports world, there is a growing demand for protein-containing products to support optimal adaptation to training [3], and elderly people are advised to eat a more protein-rich diet than young people relative to their body mass to combat sarcopenia [4]. In order to meet this dramatically increased demand in protein-rich food products, it is pivotal to investigate and identify alternative dietary protein sources with high nutritional quality.

Protein quality is related to the amino acid (AA) composition and protein digestibility [5,6]. Of the 20 AAs, nine of them are defined as essential AAs (EAAs), two are conditionally essential, and the remaining nine are considered non-essential AAs (NEAAs) [7]. EAAs cannot be synthesized in the human body and must therefore be obtained through the diet. Current research has shown that animal and dairy-based protein sources have a higher quality compared to plant-based sources due to differences in AA composition (e.g., higher amounts of EAAs and leucine), the rate of digestion and delivery to peripheral versus abdominal organs [8]. Furthermore, protein sources characterized by a relatively high content of all EAAs will reduce the need for protein per kg of body weight compared to incomplete protein sources. In addition, the ingestion of EAAs is essential for the optimization of the muscle protein synthesis rate in general and the support of muscle adaptation and growth when combined with physical training [8,9,10]. Furthermore, besides being a building block for the synthesis of proteins, the branched-chain amino acid (BCAA) leucine stimulates muscle protein synthesis [11]. Therefore, the amount and bioavailability of EAAs are crucial for supporting muscle protein mass.

Whey protein is considered a superior protein source with respect to supporting muscle growth, based on its high bioavailability of AAs, its EAA profile and its high content of leucine [12]. Similarly, soy is considered to be the best plant-based alternative, but is still inferior compared to whey protein [12]. Furthermore, soybean agriculture has a devastating impact on the environment in Asia and South America due to aspects such as deforestation, erosion and the extensive use of pesticides [13]. Therefore, in the present trial we studied krill protein and its potential to become a new protein source in human nutrition.

In a recent paper by Froehlich et al., 2018, it was estimated that if the future growth in meat consumption were substituted with seafood, the land-use savings would be equal to twice the size of India or about 740 million hectares [14]. It is therefore important to search for additional sustainable protein sources, which can be found in the form of a tiny organism from the Southern Ocean, the Antarctic krill (*Euphausia superba*). Antarctic krill is a shrimp-like crustacean with one of the world’s largest single-species biomasses. Because of a very conservative catch quota and biomass trends, krill stocks are among the best managed and most underutilized marine protein resources to date [15]. Krill meal, which is produced from whole ground krill, is a combination of high-quality protein containing all EAAs and lipids that are rich in omega-3 long-chain polyunsaturated fatty acids [16]. Because of its nutritional profile and feed attractant properties, krill meal is extensively used in aquaculture and pet feeds [17,18,19]. For humans, the benefits of supplementation of only the lipid part (krill oil) and protein/lipid mixes (krill powder/krill protein concentrate) have been investigated in various pre-clinical and clinical studies [20,21]. Some preclinical and in vitro studies were performed with a focus on krill protein alone [22,23,24,25], but supplementation of a krill protein hydrolysate (KPH) in humans is yet to be studied.

To the best of our knowledge, investigations of the blood serum AA profile after the ingestion of krill protein compared to other protein sources have not been reported to date. Consequently, the primary aim of the present study was to investigate the acute postprandial bioavailability of AAs from a new krill protein hydrolysate compared to two common and commercial protein ingredients on the market, soy protein isolate and whey protein isolate. For this purpose, serum levels of AA were studied after a protein-matched dose during a 180-min postprandial period using ^1^H NMR spectroscopy, which was previously demonstrated to be a valuable tool for the characterization of postprandial blood plasma AA profiles [26]. The krill protein used in this trial was enzymatically hydrolyzed, and we therefore expected the postprandial absorption of AAs to be faster compared to non-hydrolyzed protein isolates. The effect of hydrolyzing a protein isolate on postprandial AA bioavailability has previously been investigated in both whey, soy and collagen protein sources [27,28,29], but never in relation to krill protein in humans. In addition, to the best of our knowledge, studies investigating the effect of krill protein on appetite have not yet been reported. Therefore, our secondary aims were to study the insulinotropic response in the blood after the intake of the three protein sources and the influence of the protein servings on satiety and hunger.

## 2. Materials and Methods

### 2.1. Subjects

Ten healthy young males between 24 and 30 years of age participated in this study. Subject characteristics are provided in Table 1. The exclusion criteria were smoking, medication or health issues conflicting with the experimental trial. The trial complied with the Declaration of Helsinki and was approved by The Central Denmark Region Committees on Health Research Ethics (Journal 69,302, case number 1-10-72-155-19). The project was registered at ClinicalTrials.gov (ID NCT04864769). All subjects gave their informed consent to participate before the experiment was carried out.

### 2.2. Experimental Protocol

This study was conducted as a randomized, placebo-controlled crossover trial over four non-consecutive experimental days at the Exercise Biology Research Unit, Department for Public Health, Aarhus University, Denmark.

On the experimental days, the subjects consumed water (CON) or one of three protein matched supplements in random order (35 g of true protein (35 g (SUM of AA/100) see Table 2). The protein supplements were either whey protein isolate (WHEY) (Whey Iso, LinusPro Nutrition Aps, Risskov, Denmark); soy protein isolate (SOY) (Soy Protein, LinusPro Nutrition Aps, Risskov, Denmark); or krill protein hydrolysate (KRILL) from *Euphausia superba* (INVI^TM^ protein, Aker Biomarine Antarctic AS, Lysaker, Norway).

On the morning of the first trial, the subjects’ body composition was evaluated using whole-body dual-energy X-ray absorptiometry (DXA). During the trial, blood samples were collected as described in Section 2.5 (Figure 1).

On each experimental day, the subjects arrived at 8.00 a.m. in the laboratory in a fasted state from 10.00 p.m. the day before. A baseline blood sample was obtained, after which the subjects consumed one of three blinded drinks (WHEY, SOY or KRILL) or water (CON). All drinks were consumed within 2 min and this was designated as time 0 min. In order to improve the blinding of the products, the participants wore a nose clamp during ingestion. Blood samples and two questions, scoring based on a visual analogue scale (VAS), were collected at fixed timepoints over the following 180 min. Each subject remained in a resting position during the entire trial period. Each trial was separated by at least 48 h to ensure that the subject fully recovered between trials. The test period was completed within three weeks of the first trial.

To ensure standardization before each day, subjects recorded their dietary intake and physical activity level by completing a questionnaire the day before the first trial, which they were later instructed to follow as dietary and physical activity restrictions before the following three experimental days. All subjects were eating a mixed Danish diet, which for all subjects included a milk product and at least one other animal protein source (meat, fish, egg) during the day. Furthermore, the subjects monitored their step counts the day before each experimental day to check for comparable physical activity levels the day before testing.

### 2.3. Protein Supplement

Three types of protein powders were included in this study. The protein servings contained 35.0 g of true protein (≈135 kcal) either from commercially available whey protein isolate (35.3 g powder), commercially available soy protein isolate (39.6 g powder), krill protein hydrolysate (35.1 g powder) dissolved in 400 mL water or the placebo (400 mL water, 0 kcal) (Table 2). The krill protein was dissolved in only 100 mL water for ease of serving. After the intake of the krill protein solution, 300 mL pure water was served to standardize the servings among the products. The AA profile and mineral content were analyzed at Eurofins Steins Laboratory A/S (Vejen, Denmark). The AA content was analyzed prior to the trials, and the true protein content of the three protein sources was calculated based on the sum of the AAs (WHEY; 99%, SOY; 88% and KRILL; 99%) according to the guidelines from WHO [6]. The nutritional content and AA acid profile are presented in Appendix A.

### 2.4. Protein AA Profile and Analysis

All AA analyses of the protein supplements were performed by the Eurofins Steins Laboratory A/S (Vejen, Denmark). The AA content was determined with HPLC using a Biochrom 30+ amino acid analyzer (Biochrom, Cambridge, England) using ninhydrin reagent 440 and 570 nm for post-column derivatization. All samples were hydrolyzed for 24 h at 110 °C with 6 *N* HCL prior to AA analysis. Sulphur-containing AAs were analyzed after cold performic acid oxidation overnight and subsequent hydrolysis. Tryptophan levels were determined after alkaline hydrolysis for 22 h at 110 °C. After hydrolysis, the samples were pH-adjusted (pH 1.0–2.5), brought to volume with a loading buffer (pH 2.2) and filtered. For quantification, a one-point calibration was used for each AA, with a 0.0 comparison range to a known reference concentration. The references used were from Sigma-Aldrich, obtained from Merck Life Science (Søborg, Denmark). For quality assurance, an in-house standard, pet food, was analyzed in every run. All analyses met the criteria of EURL 152/2009 and ISO 13903:2005.

### 2.5. Blood Samples

On each experimental day, eight blood samples of ~6 mL were collected. Before the subjects consumed the protein beverage, a baseline blood sample was taken (Pre; 10 min prior to beverage consumption), after which they consumed the intervention beverage within <2 min. Seven blood samples were thereafter collected at 0, 20, 40, 60, 90, 120 and 180 min after ingestion of the protein supplementation. The blood samples were centrifuged at 1200× *g* for 10 min at 4 °C to separate the serum and were stored at −80 °C until further analysis. Serum insulin concentrations were analyzed using a Roche Cobas e601 (Roche, Mannheim, Germany) at Aarhus University Hospital, Aarhus, Denmark.

### 2.6. Serum Amino Acid Concentration Determined by ^1^H NMR Spectroscopy

Serum samples were thawed and subsequently filtered using 10 K Amicon Ultra centrifugal filter units (Merck Milipore Ltd., Cork, Ireland). A volume of 500 µL serum was added to the filter units and centrifuged for two hours at 4 °C, 14,000× *g*. In order to remove traces of glycerol, filters were washed three times with MilliQ water prior to use. A volume of 400 µL serum filtrate was transferred to a 5 mm NMR tube in addition to 100 µL phosphate buffer (50 mM Na_2_HPO_4_ in final volume, pH 7.4), 75 µL deuterium oxide (D_2_O) and 25 µL D_2_O containing 0.05% 3-(trimethylsilyl)propionic-2,2,3,3-d_4_ acid and sodium salt (TSP; Sigma-Aldrich, St. Louis, MO, USA). ^1^H NMR spectroscopy was conducted using a Bruker Avance III 600 MHz NMR spectrometer operating at a ^1^H frequency of 600.13 MHz equipped with a 5-mm ^1^H TXI probe (Bruker Biospin, Reinstetten, Germany). Spectra were acquired by applying a one-dimensional (1D) nuclear Overhauser enhancement spectroscopy (NOESY)-presat pulse sequence (noesypr1d) with pre-saturation for water suppression. The following acquisition parameters were used: target temperature = 298 K, number of scans (NS) = 128, spectral width (SW) = 7289 Hz (12.15 ppm), data points (TD) = 32,768, acquisition time (AQ) = 2.25 sand relaxation delay (D1) = 5 s. The free induction decays (FIDs) were multiplied by a line-broadening function of 0.3 Hz and were Fourier-transformed. Using TopSpin 3.0 (Bruker BioSpin, Rheinstetten, Germany), the obtained spectra were baseline- and phase-corrected. Metabolites were assigned and quantified in Chenomx NMR Suite 8.13 (Chenomx Inc., Edmonton, AB, Canada) using TSP as an internal standard.

### 2.7. Visual Analogue Scale

Two questions regarding hunger and satiety were assessed using 100-mm horizontal unbroken VAS. Subjects were asked to plot a point on the line representing their rate of, e.g., “how hungry do you feel?”, where each end point of the line stated the extremes, e.g., “not hungry at all” or “I have never been more hungry”. Each set of VAS questions was answered at every blood sampling. The intensity of the feeling (distance of the vertical mark from the origin on the left) was measured, yielding a score in the range of 0 to 100 mm.

### 2.8. Dual-Energy X-ray Absorptiometry

Body composition was measured on the day of the first trial using whole-body dual-energy X-ray absorptiometry (DXA) (GE Lunar DXA scan, GE Healthcare, Madison, WI, USA). The system’s software package (enCORE software v16.0, GE Healthcare, Madison, WI, USA) was used to determine participants’ body composition, body weight, fat mass, bone mineral content and fat- and bone-free mass. The subjects were scanned at 7.30 a.m. in a fasted state from 10:00 p.m. the night before. Before the scan, subjects had their weight and height determined. All scans were performed at the Exercise Biology Research Unit, Department for Public Health at Aarhus University.

### 2.9. Statistics

For all the AA data (total AAs, total EAAs, total NEAAs, total BCAAs and individual BCAAs), the results were analyzed via two-way ANOVA repeated measurements for analysis of changes over time and the time × treatment interaction. If interactive effects were observed, a post-hoc Tukey’s multiple comparison test between treatments for each time points was performed. In addition, the incremental area under the curve (iAUC) for all AA data was calculated via the subtraction of the results on the day the subjects ingested water and was thereafter analyzed for differences between protein supplements using a one-way ANOVA.

Data for step counts the day before the experimental days (WHEY, SOY, KRILL and CON) were analyzed using a mixed-effects model (REML) due to one missing data point for one subject on one day. Similarly, data for hunger and satiety were analyzed using a REML due to one out of eight missing data timepoints on four different experimental days.

GraphPad Prism version 8 was used for the illustration and statistical analysis of the data. Data are shown as mean ± SD if not otherwise described.

## 3. Results

Step counts for each participant were monitored the day before each experimental day to check for comparable physical activity levels the day before testing. The number of steps per day on the day before the experiment did not differ between treatment days (mean ± SEM: WHEY 7121 ± 939, SOY 5489 ± 805, KRILL 6265 ± 824 and CON 4892 ± 577 steps per day, *p* = 0.22).

### 3.1. Amino Acid Concentrations

From the ^1^H NMR spectral data, 16 AAs were identified and quantified. Aspartate, glutamate, cysteine and tryptophan could not be detected using ^1^H NMR spectroscopy.

Figure 2 shows the AA concentration over time and iAUC for total AAs, total EAAs and total NEAAs, respectively. The changes in total AAs over time differed significantly between treatments (time *p* < 0.001, time × treatment interaction *p* < 0.001). For total AAs, all of the three protein treatments significantly increased serum concentrations at all timepoints between 20–120 min compared to CON. WHEY and KRILL consumption resulted in significantly higher total AA concentrations compared to SOY between 20–60 min and 20–40 min after consumption, respectively (Figure 2A). iAUC analysis showed higher overall total AA concentrations after WHEY consumption compared to SOY (∆ WHEY vs. SOY = 76,862 ± 34,478, *p* = 0.0002) and a tendency of higher iAUC for WHEY consumption compared to KRILL (∆ WHEY vs. KRILL = 49,672 ± 59,445, *p* = 0.0631), whereas no significant difference was found between KRILL and SOY (∆ KRILL vs. SOY = 27,189 ± 49,189, *p* = 0.241) (Figure 2B).

The changes in total EAAs over time differed significantly between treatments (time *p* < 0.001, time × treatment interaction *p* < 0.001). The analysis of the total EAA concentration revealed significantly higher serum concentration from 20–60 min after WHEY and KRILL consumption compared to SOY. At 60 and 90 min after WHEY consumption, significantly higher total EAA concentrations compared to both KRILL and SOY were observed (Figure 2C). iAUC analysis of total EAAs revealed a higher overall concentration following WHEY consumption compared to KRILL (∆ WHEY vs. KRILL = 40,784 ± 30,171, *p* = 0.0053) and SOY (∆ WHEY vs. SOY = 75,067 ± 27,762, *p* < 0.0001) and a higher overall concentration for KRILL consumption compared to SOY (∆ KRILL vs. SOY = 34,283 ± 20,912, *p* = 0.0015) (Figure 2D).

The changes in total NEAAs over time differed significantly between treatments (time *p* < 0.001, time × treatment interaction *p* < 0.001, Figure 2E). The total NEAA level was significantly increased after WHEY and KRILL compared to CON after 20 min (*p* < 0.05). Furthermore, total NEAAs were significantly higher after the ingestion of KRILL versus SOY (*p* < 0.05). After 40, 60 and 90 min of ingestion of the supplement, total NEAAs were significantly increased after all three protein servings compared to CON ingestion (Figure 2E). No significant differences among the protein treatment days were found in the iAUC analysis for total NEAAs (∆ WHEY vs. SOY = 4871 ± 19,429, ∆ WHEY vs. KRILL = 7185 ± 34,364, ∆ SOY vs. KRILL = 2314 ± 34,801) (Figure 2F).

The changes in total BCAAs over time differed significantly between treatments (time *p* < 0.001, time × treatment interaction *p* < 0.001, Figure 3A). Total BCAA concentrations were significantly enhanced after the ingestion of all protein servings compared to CON ingestion after 20–180 min (*p*-values < 0.01), except CON versus WHEY at 180 min (*p* = 0.076). Total BCAAs were higher after the ingestion of KRILL than after SOY 40 min (*p* < 0.001) and 60 min (*p* = 0.0603). Furthermore, total BCAAs were higher after the ingestion of WHEY than SOY at 60 and 90 min (*p*-values < 0.01). iAUC analysis for total BCAAs revealed higher iAUC after WHEY consumption compared to KRILL (∆ WHEY vs. KRILL = 27,465 ± 20,922, *p* = 0.0063) and SOY (∆ WHEY vs. SOY = 42,451 ± 20,226, *p* = 0.0003) and higher iAUC after KRILL consumption compared to SOY (∆ KRILL vs. SOY = 14,987 ± 14,692, *p* = 0.0255) (Figure 3B).

Analysis of the specific BCAAs leucine, isoleucine and valine revealed for all three AAs that the changes in the concentration over time differed significantly between treatments (time *p* < 0.001, time × treatment interaction *p* < 0.001, Figure 3C,E,G). For leucine, KRILL consumption significantly increased leucine concentrations after 20 and 40 min compared to SOY (*p*-values < 0.05). WHEY consumption significantly increased leucine concentrations from 40–90 min compared to SOY and between 60–90 min compared to KRILL (Figure 3C). The iAUC analysis of leucine revealed increased overall leucine concentration after WHEY consumption compared to SOY and KRILL (∆ WHEY vs. SOY = 15,954 ± 6391, ∆ WHEY vs. KRILL = 12,200 ± 7409, *p*-values <0.01) and higher overall leucine concentrations after KRILL consumption compared to SOY (∆ KRILL vs. SOY = 3754 ± 4067, *p* = 0.0411) (Figure 3D). For isoleucine, the serum concentration increased significantly for KRILL compared to SOY at 20, 40 and 60 min following consumption. WHEY consumption significantly increased isoleucine concentrations compared to SOY at all timepoints between 20 and 90 min and at 60 and 90 min compared to KRILL consumption (Figure 3E). iAUC analysis for isoleucine revealed higher iAUC values after WHEY consumption compared to SOY and KRILL (∆ WHEY vs. SOY = 13,478 ± 4630, ∆ WHEY vs. KRILL = 9753 ± 5844, *p* < 0.0001 and *p* = 0.0013) and higher iAUC values after KRILL consumption compared to SOY (∆ KRILL vs. SOY = 3726 ± 4108, *p* = 0.0445) (Figure 3F). Valine was significantly enhanced after the ingestion of all protein servings compared to CON ingestion after 20 to 180 min (*p*-values < 0.01), except CON versus SOY at 20 min (*p* = 0.0678) and CON versus WHEY at 180 min (*p* = 0.0578). Valine levels were higher after the ingestion of WHEY and KRILL compared to SOY at 20, 40 and 60 min after the protein servings (*p*-values ≤ 0.05). Furthermore, the increase in valine was significantly greater after WHEY compared to SOY and KRILL protein ingestion 90 min after the protein serving (*p*-values < 0.05, Figure 3G). iAUC analysis of valine revealed significantly increased iAUC after WHEY consumption compared to SOY (∆ WHEY vs. SOY = 13,045 ± 10,464, *p* = 0.0086) and increased iAUC after KRILL consumption compared to SOY (∆ KRILL vs. SOY = 7548 ± 7691, *p* = 0.031), whereas no significant difference was found between WHEY and KRILL consumption (∆ WHEY vs. KRILL = 5497± 9152, *p* = 0.1942 (Figure 3H).

### 3.2. Insulin

The changes in serum insulin over time differed significantly between treatments (time *p* < 0.001, time × treatment interaction *p* < 0.001, Figure 4A). Serum insulin was significantly increased after ingestion of all three protein supplements compared to CON after 20, 40 and 60 min (*p* < 0.01). Furthermore, 20 min after ingestion, serum insulin was significantly higher after ingestion of KRILL versus SOY (*p* < 0.05). However, 90 min after protein ingestion, serum insulin was only significantly enhanced compared to CON after SOY (*p* < 0.01) and WHEY (*p* < 0.01), but not after ingestion of KRILL (*p* = 0.25). Serum insulin was not different after the ingestion of the protein supplements at 120 and 180 min compared to after ingestion of CON (Figure 4A). No significant differences in serum insulin among the protein supplements were found in the iAUC analysis for serum insulin (∆ WHEY vs. SOY = 1131 ± 2351, ∆ WHEY vs. KRILL = 44 ± 2480, ∆ KRILL vs. SOY = 1087 ± 2052) (Figure 4B).

### 3.3. VAS Score

During the experimental days, the subjects were asked to note hunger and satiety on a VAS. The results showed significant changes in hunger and satiety during the experimental day, but no time × treatment interaction (Figure 5).

## 4. Discussion

The global demand for protein-containing foods is increasing due to the increasing world population [2]. Since adequate protein intake is essential to humans [1], identifying new high-quality and sustainable protein sources is needed. Therefore, we investigated the acute postprandial bioavailability of AAs from a new KRILL protein hydrolysate as compared to two commercial protein sources, a SOY and a WHEY protein isolate. ^1^H NMR-based metabolomics has previously been shown to be a useful tool to determine postprandial AA concentrations in blood [26,29]. Thus, in the present study, ^1^H NMR spectroscopy was used to determine the serum AA concentrations during a 180-min postprandial period.

Studies on krill protein for human consumption are very limited [30] and the AA profile after the ingestion of krill protein compared to other protein sources has not been studied previously in a human trial. The study revealed that all of the three protein sources significantly increased the total postprandial AA serum concentration at all timepoints between 20–120 min compared to CON. However, WHEY and KRILL consumption resulted in significantly higher total AA concentration compared to SOY between 20–60 min and 20–40 min after consumption, respectively. iAUC analysis revealed a significantly increased total serum AA concentration following WHEY consumption compared to SOY and a tendency of an increased total AA concentration following WHEY consumption compared to KRILL (*p* = 0.0631). The fact that WHEY consumption resulted in a higher postprandial AA concentration compared to SOY is in agreement with previous studies demonstrating higher postprandial AA concentrations in the blood after whey consumption compared to soy [26,27,28].

From the iAUC analyses, it was evident that WHEY consumption resulted in the highest postprandial serum concentrations of EAAs and BCAAs, followed by KRILL and SOY consumption, respectively. It was also found that postprandial serum concentrations of EAAs and BCAAs were higher following KRILL consumption when compared with SOY. No significant differences among the treatment groups were found in the iAUC analysis of NEAAs. The presence of EAAs is important for muscle protein synthesis, and studies have indicated that increasing postprandial EAA concentrations in the blood can stimulate MPS regardless of the intake of NEAA [9,10,31]. The increased postprandial concentrations of EAAs and BCAAs following WHEY consumption compared to SOY were in accordance with previous studies [26,27,28], and whey consumption has previously been shown to stimulate MPS to a greater extent than soy and casein [28]. Based on the findings of the previous studies, it could be speculated that krill protein hydrolysate is superior to soy in stimulating MPS, which may at least partly be related to the fact that the krill protein was extensively hydrolyzed (degree of hydrolysis ~34–35). Studies have indicated that BCAAs and leucine in particular play a key role in the stimulation of MPS [11,32]. In the present study, analysis of the specific BCAAs revealed that WHEY consumption resulted in the highest leucine and isoleucine concentrations, followed by KRILL and SOY, respectively, and valine concentrations were increased after WHEY and KRILL consumption compared to SOY. The higher postprandial leucine concentrations following WHEY consumption compared to SOY were in accordance with previous findings [26,28]. The higher overall postprandial serum concentration of leucine following WHEY consumption compared to KRILL is in accordance with the fact that the WHEY protein isolate has a higher content of leucine compared to the KRILL protein hydrolysate. However, the postprandial serum concentrations of leucine at 0–40 min following consumption did not differ between the WHEY and KRILL protein treatment day. The latter may be explained by the fact that the KRILL supplement was hydrolysed, which may have had a positive influence on the digestibility of the protein and thereby the absorption of leucine. Leucine is among the AAs that are well known to stimulate insulin secretion [33]. As expected, the insulin response increased following the consumption of all of the protein treatments compared to CON. At 20 min following consumption, KRILL consumption resulted in significantly increased insulin concentrations compared to SOY. Hence, the increased insulin response following KRILL consumption at 20 min post-consumption might be explained by the significantly higher serum leucine concentration following KRILL consumption compared to SOY. However, it should be noted that iAUC analyses did not reveal overall significant differences in insulin responses among the protein treatment groups. In general, protein hydrolysate contains mostly di- and tripeptides, which are more rapidly absorbed than longer peptides [27,34,35]. Still, the hydrolysis of the krill protein did not provide an advantage over the non-hydrolyzed WPI, when aminoacidemia patterns were compared. We are aware that the AA composition also differed between the two supplements, but our observation is in line with those of Calbet et al., who found no difference in the pattern of aminoacidemia after the consumption of 36 g of whole whey protein versus hydrolyzed whey protein [36]. As did other studies using 25 g [37] and 45 g [38] of either intact or hydrolyzed whey protein. In these examples, it seems that intact whey is already efficiently absorbed with similar absorption kinetics to hydrolyzed proteins. However, it is not only the degree of hydrolyzation that determines absorption kinetics. As an example, it has been shown that the transepithelial transport pathway of casein-derived peptides is affected by the molecular weight of the peptides [39]. Hence, it could be speculated that not only the degree of hydrolyzation, but also the pathway used for transepithelial transport, affects the bioavailability of a given peptide. Furthermore, independently of absorption kinetics, hydrolyzed protein may have beneficial physiological effects. In a study with rats, it was shown that hydrolyzed whey protein stimulated MPS more efficiently than intact whey protein, even though lower aminoacidemia was observed after ingestion of the hydrolysate [40]. This finding is possibly related to a higher plasma insulin concentration observed after ingestion of the whey protein hydrolysate compared to the intact whey protein, since insulin may support MPS when the AA availability is sufficient [41]. Hence, it may be of interest for future studies to measure the effect of the ingestion of krill protein hydrolysate on muscle protein fractional synthetic rates at rest and after exercise, since in the present study an initial comparable increase in insulin after KRILL and WHEY was observed.

Protein intake has been shown to regulate food intake and a protein-rich diet has been suggested to induce satiety and suppress food intake to a greater extent than a diet rich in fat or carbohydrates [42,43,44]. Furthermore, the protein source seems to affect food intake [45,46]. In the present study, differences in satiety and hunger were observed over time during the test day. The mechanisms of the satiating effects of proteins are elusive, but one of the proposed mechanisms is the so-called aminostatic hypothesis, suggesting that elevated AA concentrations in blood act as a satiety signal [47]. However, despite differences in postprandial serum AA concentrations between the treatment groups, no time × treatment interactions for satiety or hunger were observed. Veldhorst et al., (2009) found that for a meal with a high protein content (25E%), there was no difference in appetite sensations between whey, casein and soy [48]. Thus, the lack of differences in satiety in the present study might be explained by the relative high protein dose, meaning that for all three protein sources, blood AA concentrations were above a certain threshold and thus provided a satiating effect.

In terms of possible limitations to this study, the sample size of 10 participants is relatively small. In addition, our findings apply only to a young healthy population at rest. Future studies are warranted including elderly participants, who may absorb and respond to KRILL supplements differently compared to young subjects [49]. In addition, since the study only included healthy young men, the effect of this intervention in women is unknown. It will also be of interest to study the acute interactive effect of krill protein ingestion and exercise on muscle protein turnover, as well as on muscle growth after a prolonged intervention.

## 5. Conclusions

The study revealed that the consumption of a KRILL protein hydrolysate significantly increased the postprandial serum AA concentrations 20–120 min post-consumption. Comparison with WHEY isolate and SOY protein isolate ingestion revealed that the KRILL hydrolysate resulted in higher postprandial concentrations of EAAs and BCAAs than SOY, whereas WHEY ingestion induced the highest postprandial concentrations of EAAs and BCAAs. Thus, the study indicated that KRILL protein hydrolysate is superior to SOY protein isolate in increasing postprandial serum EAA and BCAA concentrations and might therefore represent a promising future protein source. Still, our data underlined the nutritional high value of whey protein with respect to amino acid profiles and digestibility.

## Figures and Tables

**Figure 1 nutrients-13-03187-f001:**
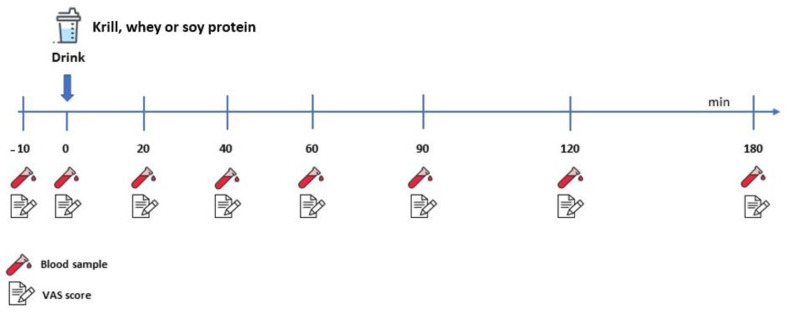
Experimental protocol.

**Figure 2 nutrients-13-03187-f002:**
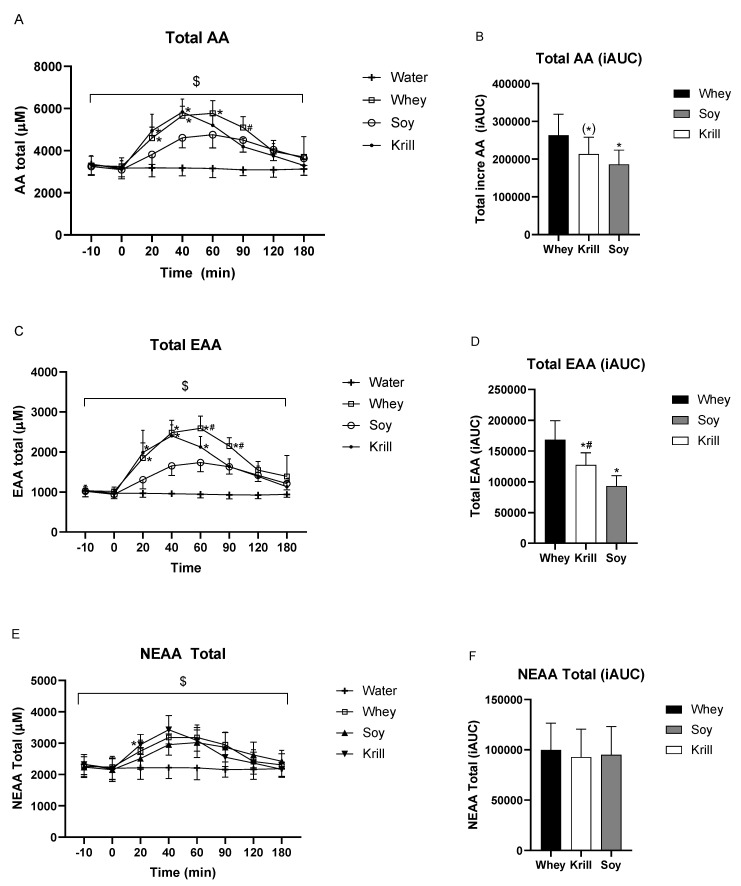
Postprandial serum concentrations of total (**A**) amino acids (AAs), (**C**) essential amino acids (EAAs) and (**E**) non-essential amino acids (NEAAs) and incremental area under the curve (iAUC) analyses for total (**B**) AAs, (**D**) EAAs and (**F**) NEAAs. $ *p* < 0.001 time × treatment interaction. For Figure 2A,C,E: * significantly different from SOY, # significantly different from Krill. For Figure 2B,D: (*) *p* = 0.0631 compared to WHEY, * *p* < 0.05 compared to WHEY, # *p* < 0.05 KRILL compared to SOY.

**Figure 3 nutrients-13-03187-f003:**
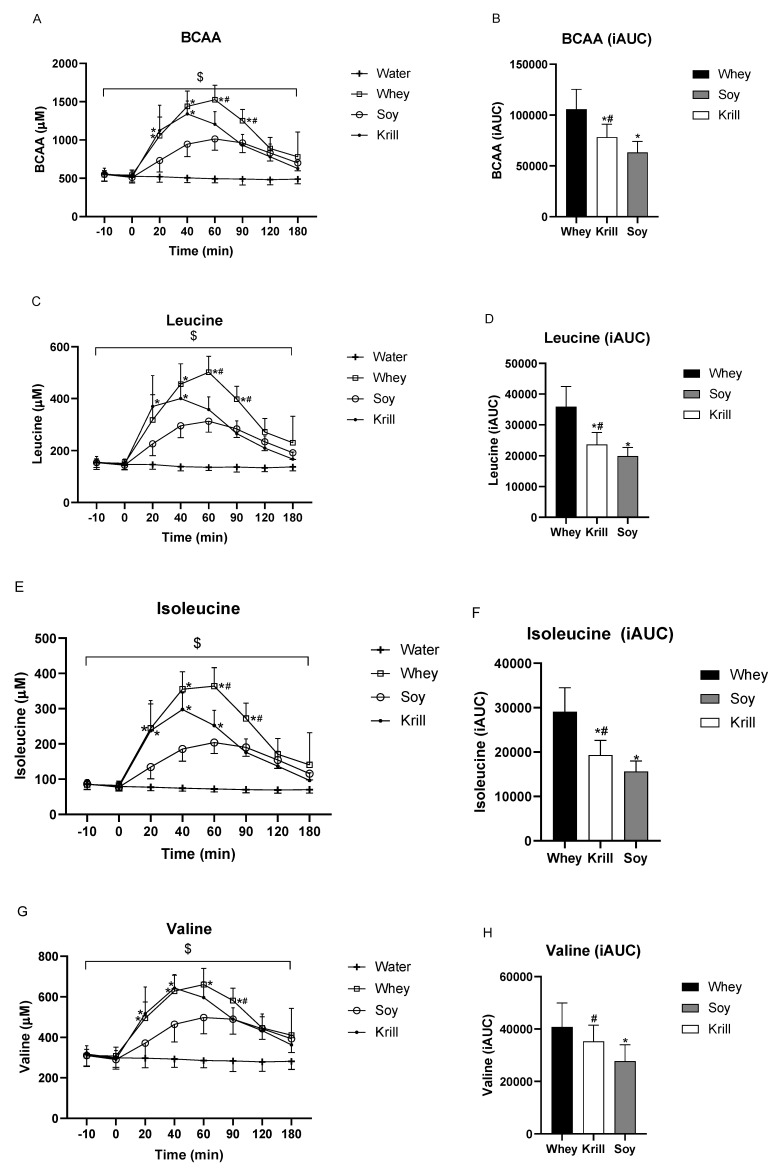
Postprandial serum concentrations of total (**A**) branched chain amino acids (BCAAs), (**C**) leucine, (**E**) isoleucine, (**G**) valine and incremental area under the curve (iAUC) analyses for total (**B**) BCAAs, (**D**) leucine, (**F**) isoleucine and (**H**) valine. $ *p* < 0.001 time × treatment interaction. For Figure 3A,C,E,G: * significantly different from SOY, # significantly different from Krill. For Figure 3B,D,F,H: * *p* < 0.05 compared to WHEY, # *p* < 0.05 KRILL compared to SOY.

**Figure 4 nutrients-13-03187-f004:**
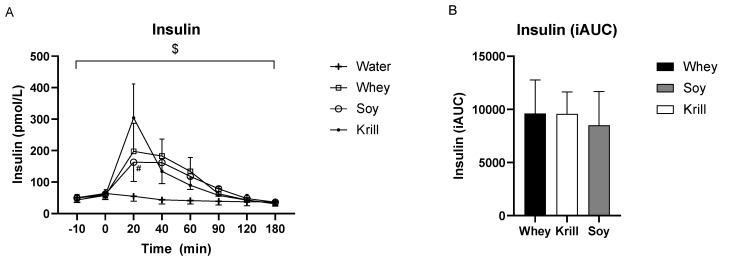
(**A**) Serum insulin concentration over time and (**B**) area under the curve (iAUC) analyses for serum insulin concentration. $ *p* < 0.001 time × treatment interaction. # Significantly different from KRILL.

**Figure 5 nutrients-13-03187-f005:**
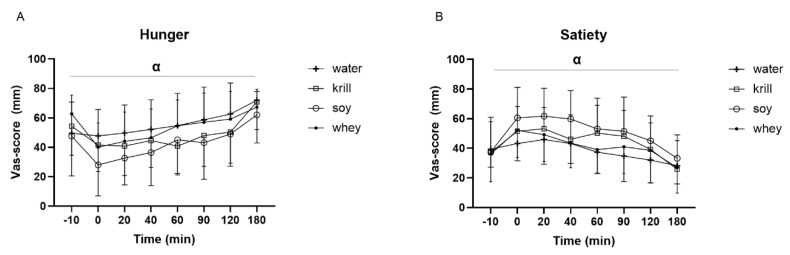
(**A**) Hunger and (**B**) satiety determined on a VAS-score scale over time. α *p* < 0.05 for significant change over time. No significant (NS) time × treatment interaction.

**Table 1 nutrients-13-03187-t001:** Subject characteristics.

	Subjects (*n* = 10 Men)	
Age (years)	27	±2
Weight (kg)	81.5	±10.0
Height (cm)	186	±6
BMI (kg/m^2^)	23.6	±2.7
Activity level (hours/week)	3.5	±2.4

**Table 2 nutrients-13-03187-t002:** Amino acid content in test drinks.

	Krill Protein Hydrolysate (35.1 g Powder)	Whey Protein Isolate (35.3 g Powder)	Soy Protein Isolate (39.6 g Powder)
Macronutrient composition			
Energy (kcal)	132	130	155
Protein (g) ^¤^	32.7	30.7	34.8
Fat (g)	<0.4	0.3	1.5
Carbohydrates (g)	0.0	1.1	0.0
Alanine, g	2.02	1.83	1.47
Arginine, g	2.17	0.67	2.54
Asparagine, g	4.08	3.78	4.20
Cysteine, g	0.27	0.76	0.40
Glutamine, g	5.20	6.17	6.62
Glycine, g	1.57	0.48	1.44
Histidine, g *	0.85	0.52	0.91
Isoleucine, g *^,#^	1.77	2.17	1.57
Leucine, g *^,#^	2.87	3.63	2.73
Lysine, g *	3.28	3.26	2.16
Methionine, g *	1.02	0.82	0.47
Phenylalanine, g *	1.62	0.97	1.81
Proline, g	1.27	2.18	1.84
Serine, g	1.44	1.62	1.86
Threonine, g *	1.75	2.55	1.38
Tyrosine, g *	1.52	1.02	1.44
Valine, g *^,#^	1.89	1.98	1.65
Tryptophan, g	0.42	0.59	0.50
TAA	35.00	35.00	35.00
EAA *	16.56	16.91	14.13
NEAA	18.44	18.09	20.87
BCAA ^#^	6.53	7.78	5.94

^¤^ Protein content evaluated using the Kjeldahl method (N*6.25), (N*6.38 for whey protein); * essential amino acids; ^#^ branched-chain amino acids; TAA, total amino acids (true protein: sum of all amino acids); EAA, total essential amino acids; NEAA, total non-essential amino acids; BCAA, total branched-chain amino acids.

## Data Availability

Data will be available by contacting the corresponding author up to five years after publication. However, before reuse an acceptance from the participants for reuse is mandatory.

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
