# Peer review of "Krill Protein Hydrolysate Provides High Absorption Rate for All Essential Amino Acids—A Randomized Control Cross-Over Trial"

_nutrients, 2021, doi:10.3390/nu13093187_

Round 1

Reviewer 1 Report

Comments to the Author

This is a practically important research indicates that the krill protein hydrolysate provides high absorption rate for all essential amino acids – a randomized control cross-over trial

However, it is necessary to reexamine the research method etc. in several respects.

1.Background

Secondary aims were to study the insulinotropic response in the blood after intake of the three protein sources and the influence of the protein servings on satiety and hunger.

What previous studies have been reported on krill protein intake and appetite suppression?

The reasons for investigated of satiety and hunger should be described in the background.

2. Method

The sample size of 12 people is small.

Please describe your rationale for setting the sample size in this study.

3. Method

Were subjects who habitually performed in resistance exercise and intake protein supplements (whey or soy protein supplements) included before the experimental trial?

Before the experimental trial, subjects with a history of resistance exercise and protein supplementation should have been excluded.

4. Method

Describe in detail any dietary and physical activity restrictions on the subject.

5. Results

To ensure standardization before each day, subjects recorded dietary intake and physical activity level by a questionnaire the day before the first trial.

Step counts for each participant were monitered the day before each experimental day to check for comparable physical activity level the day before testing. Number of steps per day the day before the experiment did not differ between treatment days.

In Table 1, describe the energy and nutrition (animal protein etc.) intake amount of the subjects on the day before the first experiment trial.

6. Discussion

In this study indicated that krill protein hydrolysate is superior to soy protein isolate in increasing postprandial serum EAA and BCAA concentrations and might therefore be a promising future protein source.

However, at 60 and 90 min after krill consumption, significantly lower total EAA concentrations compared to whey were observed (Figure 2C). AUC analysis of total EAA revealed a lower overall concentration following krill consumption compared to whey (Figure 2D).

It would be better to discuss the therapeutic strategy considering difference of effect between whey protein and krill protein.

7. Discussion

Add the following to the limitation.

Changes in women are unknown.

Sample size is small.

Author Response

Dear Editor and reviewers

We are very thankful for your relevant and constructive comments and suggestions. In light of the comments and suggestions we have revised the manuscript. It is our expression that this has improved the quality of manuscript. Below are point-to-point replies to the questions raised by the reviewers. We have attached the revised manuscript with red-marks showing the changes.

Reviewer 1

Comments to the Author: This is a practically important research indicates that the krill protein hydrolysate provides high absorption rate for all essential amino acids – a randomized control cross-over trial. However, it is necessary to reexamine the research method etc. in several respects.

1.Background

Secondary aims were to study the insulinotropic response in the blood after intake of the three protein sources and the influence of the protein servings on satiety and hunger.

What previous studies have been reported on krill protein intake and appetite suppression?

The reasons for investigated of satiety and hunger should be described in the background.

To the best of our knowledge, studies investigating krill protein intake and appetite/hunger have not been reported yet. Since we wanted to investigate the potential of krill as a new protein source for human consumption, we found it relevant to include satiety and hunger in the investigation, since studies covering this aspect are lacking and as there in general is a great interest in understanding the satiety potential of protein sources (Kohanmoo, Faghih, & Akhlaghi, 2020). Furthermore, in a previous resistance training project with insect protein supplementation (PRO) versus isocaloric carbohydrate supplementation (CHO) we observed an increase in energy intake related to the training intervention in the CHO group, but not in the PRO (Vangsoe, Joergensen, Heckmann, & Hansen, 2018). In further support for the PRO supplement influenced the appetite regulation and thereby the energy balance, the PRO group reduced fat mass numerically by 0.4 kg after 8 weeks, whereas the CHO group experienced a numeric increase in fat mass by 0.2 kg (Vangsoe, Joergensen, Heckmann, & Hansen, 2018). Based on the latter observations, we thought it would be relevant to include questions about satiety.

  1. Method

The sample size of 10 people is small.

Please describe your rationale for setting the sample size in this study.

We have in a previous randomized controlled cross-over trial detected a significant difference in amino acid profile after ingestion of 25 g whey protein isolate compared to insect protein isolate and soy protein isolate (Vangsoe, Thogersen, Bertram, Heckmann, & Hansen, 2018). In the latter trial we included six healthy young men (Vangsoe, Thogersen, Bertram, Heckmann, & Hansen, 2018) as in the present trial. We expected that six would be sufficient to detect a difference between Whey and Soy in amino acid profile. However, we chose to enhance the sample size to 10 to improve the statistical power for detecting a difference between WHEY and KRILL since the krill protein source was hydrolyzed which was hypothesized to reduce the difference in amino acid absorption rate between the products.

  1. Method

Were subjects who habitually performed in resistance exercise and intake protein supplements (whey or soy protein supplements) included before the experimental trial?

A history of resistance exercise and protein supplementation were not exclusion criteria since the subjects were their own control. As written in the manuscript we controlled the diet and physical activity level the day before each experimental day: “To ensure standardization before each day, subjects recorded dietary intake and physical activity level by a questionnaire the day before the first trial, which they were later instructed to follow as dietary and physical activity restrictions before the following three experimental days. Furthermore, the subjects monitored step counts the day before each experimental day to check for comparable physical activity level the day before testing.”

  1. Method

Describe in detail any dietary and physical activity restrictions on the subject.

The participants were instructed to follow their normal dietary pattern and physical activity level, but not to perform any strenuous exercise bout the day before the experimental days. As described in the manuscript, participants recorded dietary intake and physical activity level by a questionnaire the day before the first trial, to ensure standardization before each day. Later, they were instructed to follow this as dietary and physical activity restrictions before the following three experimental days. Furthermore, the subjects monitored step counts the day before each experimental day to check for comparable physical activity level the day before testing.

  1. Results

To ensure standardization before each day, subjects recorded dietary intake and physical activity level by a questionnaire the day before the first trial.

Step counts for each participant were monitered the day before each experimental day to check for comparable physical activity level the day before testing. Number of steps per day the day before the experiment did not differ between treatment days.

In Table 1, describe the energy and nutrition (animal protein etc.) intake amount of the subjects on the day before the first experiment trial.

We have the participants’ diet registration on the day before the experimental days. All subjects were eating a mixed Danish diet which for all subjects included milk product and at least one other animal protein source (meat, fish, egg) during the day. This information has been added to the revised manuscript, lines 145-147. We have not calculated the exact energy content and protein content since one day diet registration is not expected to be representative for habitual intake. Nevertheless, if the reviewer want us to calculate the energy and protein intake we have the raw data from the registration to do so.

  1. Discussion

In this study indicated that krill protein hydrolysate is superior to soy protein isolate in increasing postprandial serum EAA and BCAA concentrations and might therefore be a promising future protein source. However, at 60 and 90 min after krill consumption, significantly lower total EAA concentrations compared to whey were observed (Figure 2C). AUC analysis of total EAA revealed a lower overall concentration following krill consumption compared to whey (Figure 2D). It would be better to discuss the therapeutic strategy considering difference of effect between whey protein and krill protein.

As correctly stated by the reviewer, we found that krill protein resulted in a higher postprandial serum EAA and BCAA than soy protein whereas whey protein resulted in higher levels than krill protein. The objective of the study was to elucidate the nutritional quality in terms of amino acid bioavailability, which is the reason for reporting postprandial serum EAA and BCAA. We are not certain what the reviewer means with ‘therapeutic’ use, but we have added the following sentence:

Conclusion (abstract) “Krill protein hydrolysate increases postprandial serum EAA and BCAA concentrations superior to soy protein isolate and might be a promising future protein source in human nutrition.”

Line 470-471 (…and might therefore be a promising future protein source. Still, our data underlined the nutritional high value whey protein in respect to amino acid profile and digestibility.

  1. Discussion

Add the following to the limitation.

Changes in women are unknown.

Sample size is small.

According to reviewer’s comment, the suggested limitations have been added to the discussion (line 454-455 and 457-459 in the revised manuscript).

Reviewer 2 Report

The study provided useful information about the absorption rate for all essential amino acids of Krill protein in comparison with other two protein (whey protein isolate and soy bean protein isolate). I recommend that the manuscript be accepted. 

I have a small comment that some words in the abstract such as AUC, EAA, BCAA should be in their full forms before giving their abbreviations.

Author Response

Dear reviewer

Thank you for your positive comments to our manuscript. A reply to your comments is added below. Furthermore, we have attached the revised manuscript showing changes with red:

Comments from reviewer: The study provided useful information about the absorption rate for all essential amino acids of Krill protein in comparison with other two protein (whey protein isolate and soy bean protein isolate). I recommend that the manuscript be accepted. 

I have a small comment that some words in the abstract such as AUC, EAA, BCAA should be in their full forms before giving their abbreviations.

According to reviewer’s comment, the full forms of the abbreviations have been added to the abstract in the revised manuscript.

Reviewer 3 Report

In this cross-over intervention study 3 different protein sources were evaluated for their postprandial AA response when given in equal amounts, and compared to a water only condition. The study was carefully executed, and I in particular appreciate the standardisation of diet and physically activity behaviour the day preceding the experimental trial days.
Krill protein hydrolysate intake results in clear postprandial (E)AA response, better than soy protein isolate, lower than whey protein hydrolysate

A main issue is the comparison of a hydrolysate vs isolates. What is the degree of hydrolysis of the krill protein?

A hydrolysate will result in a more favourable AA digestion, absorption and postprandial AA profile. This is  acknowledged by the authors at several occasions. However in line 387 the authors state that ‘it could be speculated that krill protein is superior to soy in stimulating MPS’, this however not warranted, as krill protein is not investigated, neither MPS. A sentence like ‘it could be speculated that krill protein hydrolysate is superior to soy protein isolate in stimulating MPS’ reflects more their observations.
Moreover, what can be expected for Krill (non-hydrolyzed) protein? As the authors like to conclude that  it (line 460) ‘..might therefore be a promising future protein source.’

A second issue is related to the satiety (VAS) results. No time x treatment interaction is observed. So, protein conditions do not differ to water, as well as no differences between protein sources. This observation, which contrasts the general observed satiating effect of protein, is not addressed in their discussion (lines 433-446), which exclusively focuses on the between protein differences. So, what explains the lack of difference between the protein and the control condition?

Other remarks

Intervention/methods

Why was 35 g protein chosen? As an optimal meal dose is believed to be ~ 20-25 gram of protein

How was blinding done? The krill protein was dissolved in only 100 mL water to ease the serving - likely because of taste?- with additional water. So the appearance should have been different?. The water only condition was also blinded?
So, how well was blinding achieved?

Was the order of treatments randomized?

For clarity I would advise to use ‘iAUC’ (incremental AUC) as the better abbreviation throughout the text/figures

There is some repetition in the methods section, e.g. the blood sampling scheme is already present in section 2.2 and Fig 1, and again repeated in section 2.5. Check the methods section for some more repetition

For clarity rename the heading of section 2.4 to ‘Protein AA profile ..’. (in contrast to serum AA profile in section 2.6)

‘CON’ , ‘placebo’ and ‘water’ are all used to describe the water only condition. Use consistency throughout the manuscript for clarity

Table 2 versus Table 3: these contain merely the same information. Is the information in Table recalculated form the measured AA profile in Table 3?. Actually only te information in Table is relevant for this manuscript, as this is what the participants consumed. I would suggest to leave out Table 3, or make it supplementary?

Results

Lines 239-244: Was dietary behavior they day before the test days monitored (as was physical activity with the step counter); or was dietary intake only recorded on the first day. Thus, how well was dietary standardization?

Please include also the p-value for the krill soy comparison in line 258

Lines 270-271 and Fig 2E: add significance level ($) for NEAA to Figure 2E

Figure 2 (and 3 and 4): would it be nicer to visualize the post hoc differences in the figures ACE in stead of the overall ANOVA p-value ($). The latter is not surprising given the water only condition, and the main research question is on the protein comparison. The ANOVA result is still described in the text.

Figure 5: symbols are not similar (* and α)

Author Response

Dear Reviewer

We are very thankful for your relevant and constructive comments and suggestions. In light of the comments and suggestions we have revised the manuscript. It is our expression that this has improved the quality of manuscript. Below are point-to-point replies to the questions raised by the reviewers. Attached is the revised manuscript. Changes are shown with red.

Comments and Suggestions for Authors:

In this cross-over intervention study 3 different protein sources were evaluated for their postprandial AA response when given in equal amounts, and compared to a water only condition. The study was carefully executed, and I in particular appreciate the standardisation of diet and physically activity behaviour the day preceding the experimental trial days.
Krill protein hydrolysate intake results in clear postprandial (E)AA response, better than soy protein isolate, lower than whey protein hydrolysate

A main issue is the comparison of a hydrolysate vs isolates. What is the degree of hydrolysis of the krill protein?

A hydrolysate will result in a more favourable AA digestion, absorption and postprandial AA profile. This is acknowledged by the authors at several occasions. However in line 387 the authors state that ‘it could be speculated that krill protein is superior to soy in stimulating MPS’, this however not warranted, as krill protein is not investigated, neither MPS. A sentence like ‘it could be speculated that krill protein hydrolysate is superior to soy protein isolate in stimulating MPS’ reflects more their observations.
Moreover, what can be expected for Krill (non-hydrolyzed) protein? As the authors like to conclude that it (line 460) ‘..might therefore be a promising future protein source.’

According to the reviewer’s comment, it has been emphasized that the difference between effects of SOY and KRILL may be caused by the fact that KRILL was extensively hydrolyzed (degree of hydrolysis ~34-35) (Lines 393-395 in the revised manuscript).

A second issue is related to the satiety (VAS) results. No time x treatment interaction is observed. So, protein conditions do not differ to water, as well as no differences between protein sources. This observation, which contrasts the general observed satiating effect of protein, is not addressed in their discussion (lines 433-446), which exclusively focuses on the between protein differences. So, what explains the lack of difference between the protein and the control condition?

The feeling of hunger and satiety were numerically higher and lower, respectively, at the control day (water ingestion) compared to the experimental days with protein ingestion. Nevertheless, no overall significant difference or time x treatment interaction in satiety and hunger was observed. This may be explained by the fact that the size of the test meals was small (approximately 695 kJ) and the variation in reporting these parameters was rather large between subjects.

Other remarks

Intervention/methods

Why was 35 g protein chosen? As an optimal meal dose is believed to be ~ 20-25 gram of protein

Initially, the plan was to include elderly men in perspective to counteract muscle protein breakdown. In light of this, we chose to serve 35 g of protein since this is approximately the dose needed for elderly men to maxime muscle protein synthesis (0.4 g protein/kg) (Moore, Churchward-Venne, Witard, Breen, Burd, Tipton, et al., 2014). Nevertheless, to reduce the risk related to testing we for ethical reasons chose to invite young healthy men, but maintain the high dose to enhance the change for detecting a difference in amino acid profile.

How was blinding done? The krill protein was dissolved in only 100 mL water to ease the serving - likely because of taste?- with additional water. So the appearance should have been different?. The water only condition was also blinded?
So, how well was blinding achieved?

Since the treatments differed e.g. in smell and taste (in particular when comparing water and the protein supplements), completely blinding was not possible. In order to improve blinding of the products, the participants wore a nose clamp during the ingestion of the beverage. The latter fact has been added to the manuscript (Line 136-137 in the revised manuscript). The participants were non-systematically asked if they could guess which protein source they ingested. Their response rate seemed random and not completely systematically correct, which may be related to the participants wore a nose clamp during the ingestion of the beverage. Based on pre-testing we are pretty sure that the participants would have guessed the day of ingesting KRILL if they had not worn a nose clamp.

Was the order of treatments randomized?

Yes, as mentioned in the manuscript (Line 113) the order of treatments was randomized.

For clarity I would advise to use ‘iAUC’ (incremental AUC) as the better abbreviation throughout the text/figures

AUC has been changes to iAUC throughout the text and figures.

There is some repetition in the methods section, e.g. the blood sampling scheme is already present in section 2.2 and Fig 1, and again repeated in section 2.5. Check the methods section for some more repetition

According to reviewer’s comment, repetitions in the methods section have been deleted.

For clarity rename the heading of section 2.4 to ‘Protein AA profile ..’. (in contrast to serum AA profile in section 2.6)

Has been changed.

‘CON’ , ‘placebo’ and ‘water’ are all used to describe the water only condition. Use consistency throughout the manuscript for clarity

The term “CON” is now used to describe the water only condition throughout the manuscript.

Table 2 versus Table 3: these contain merely the same information. Is the information in Table recalculated form the measured AA profile in Table 3?. Actually only te information in Table is relevant for this manuscript, as this is what the participants consumed. I would suggest to leave out Table 3, or make it supplementary?

We have changed Table 3 to a supplementary table

Results

Lines 239-244: Was dietary behavior they day before the test days monitored (as was physical activity with the step counter); or was dietary intake only recorded on the first day. Thus, how well was dietary standardization?

The participants were instructed to follow their normal dietary pattern and physical activity level, but not perform any strenuous exercise bout the day before the experimental days. As described in the manuscript, participants recorded dietary intake and physical activity level by a questionnaire the day before the first trial, to ensure standardization before each day. Later, they were instructed to follow this as dietary and physical activity restrictions before the following three experimental days. Furthermore, the subjects monitored step counts the day before each experimental day to check for comparable physical activity level the day before testing.

Please include also the p-value for the krill soy comparison in line 258

The p-value has been inserted (Line 264 in the revised manuscript).

Lines 270-271 and Fig 2E: add significance level ($) for NEAA to Figure 2E

Significance level for NEAA has been inserted.

Figure 2 (and 3 and 4): would it be nicer to visualize the post hoc differences in the figures ACE in stead of the overall ANOVA p-value ($). The latter is not surprising given the water only condition, and the main research question is on the protein comparison. The ANOVA result is still described in the text.

According to reviewer’s comment, significant differences between the protein supplements have been inserted in Figure 2, 3 and 4.

Figure 5: symbols are not similar (* and α)

“*” has been changed to “α”.

References

Kohanmoo, A., Faghih, S., & Akhlaghi, M. (2020). Effect of short- and long-term protein consumption on appetite and appetite-regulating gastrointestinal hormones, a systematic review and meta-analysis of randomized controlled trials. Physiol Behav, 226, 113123.

Moore, D. R., Churchward-Venne, T. A., Witard, O., Breen, L., Burd, N. A., Tipton, K. D., & Phillips, S. M. (2014). Protein Ingestion to Stimulate Myofibrillar Protein Synthesis Requires Greater Relative Protein Intakes in Healthy Older Versus Younger Men. The Journals of Gerontology: Series A, 70(1), 57-62.

Vangsoe, M. T., Joergensen, M. S., Heckmann, L.-H. L., & Hansen, M. (2018). Effects of Insect Protein Supplementation during Resistance Training on Changes in Muscle Mass and Strength in Young Men. 10(3), 335.

Vangsoe, M. T., Thogersen, R., Bertram, H. C., Heckmann, L.-H. L., & Hansen, M. (2018). Ingestion of Insect Protein Isolate Enhances Blood Amino Acid Concentrations Similar to Soy Protein in A Human Trial. Nutrients, 10(10), 1357.

Round 2

Reviewer 1 Report

I agree with your reply.